# Sarcopenia Index as a Predictor of Clinical Outcomes in Older Patients with Coronary Artery Disease

**DOI:** 10.3390/jcm9103121

**Published:** 2020-09-27

**Authors:** Hak Seung Lee, Kyung Woo Park, Jeehoon Kang, You-Jeong Ki, Mineok Chang, Jung-Kyu Han, Han-Mo Yang, Hyun-Jae Kang, Bon-Kwon Koo, Hyo-Soo Kim

**Affiliations:** Department of Internal Medicine and Cardiovascular Center, Seoul National University Hospital, Seoul 03080, Korea; cardiolee@gmail.com (H.S.L.); medikang@gmail.com (J.K.); drkiyou@gmail.com (Y.-J.K.); oklizard81@gmail.com (M.C.); hpcrates@gmail.com (J.-K.H.); hanname@gmail.com (H.-M.Y.); nowkang@snu.ac.kr (H.-J.K.); bkkoo@snu.ac.kr (B.-K.K.); hyosoo@snu.ac.kr (H.-S.K.)

**Keywords:** percutaneous coronary intervention, sarcopenia, creatinine, cystatin C

## Abstract

To demonstrate the association of the serum creatinine/serum cystatin C ratio (sarcopenia index, SI) with clinical outcomes including cardiovascular and bleeding risk in older patients who underwent percutaneous coronary intervention (PCI), we analyzed a multicenter nation-wide pooled registry. A total of 1086 older patients (65 years or older) who underwent PCI with second-generation drug-eluting stents (DES) were enrolled. The total population was divided into quartiles according to the SI, stratified by sex. The primary clinical outcomes were major adverse cardiovascular events (MACE, all-cause death, myocardial infarction and target lesion revascularization) and thrombolysis in myocardial infarction major and minor bleeding during a 3-year follow-up period. In the total population, MACE occurred within 3 years in 154 (14.2%) patients. The lowest SI quartile group (Q1) had a significantly higher 3-year MACE rate (Q1 vs. Q2–4; 23.1% vs. 11.2%, *p* < 0.001), while bleeding event rates were similar between the groups (Q1 vs. Q2–4; 2.6% vs. 2.2%, *p* = 0.656). The Cox proportional hazard model showed that lower SI is an independent predictor for MACE events (HR 2.23, 95% CI 1.62–3.07, *p* < 0.001). The SI, a surrogate for the degree of muscle mass, is associated with cardiovascular and non-cardiovascular death, but not with bleeding in older patients who underwent PCI.

## 1. Introduction

Sarcopenia is a progressive, generalized skeletal muscle disorder characterized by low muscle strength, low muscle quantity or quality, and low physical performance [1]. The conventional diagnostic criteria for sarcopenia include both low muscle mass and low muscle function. However, it is not easy to accurately measure muscle mass in routine clinical practice, because measuring the skeletal muscle index requires bioimpedance analysis, computed tomography (CT), dual energy X-ray absorptiometry (DEXA) or magnetic resonance imaging (MRI) [2]. By contrast, an indicator that can be easily calculated that reflects skeletal muscle simply is the serum creatinine/serum cystatin C ratio, the so-called sarcopenia index (SI) [3,4,5,6]. Studies have also been published suggesting that low SI can predict poor prognosis in severely ill patients or those with cancer [5,6,7].

However, the clinical relevance of SI in older patients undergoing percutaneous coronary intervention (PCI) who have low skeletal muscle or sarcopenia is limited and insufficient [8,9]. Therefore, the purpose of this study was to demonstrate the association of SI with clinical outcomes including cardiovascular and bleeding risk in older patients who underwent PCI.

## 2. Methods

### 2.1. Study Population

This study is based on the Grand drug-eluting stents (DES) registry, which was introduced in previous studies [10]. The Grand DES registry is a multicenter nation-wide pooled registry of the EXCELLENT registry, HOST-PRIME registry, RESOLUTE-Korea registry, RESOLINTE registry, and HOST-BIOLIMUS registry in Korea (NCT03507205). From January 2005 to November 2014, a total of 17,286 patients were enrolled from 55 centers in Korea. Clinical follow-up was performed for three years after index PCI. For analysis we only included the patients with 2nd-generation drug-eluting stents (DES) implantations. 

Among the total population, 1086 patients who were over 65 years old, and had serum creatinine and cystatin C data available were included in this study (Figure 1). The study complied with the provisions of the Declaration of Helsinki, and the study was approved by the institutional review board at each center. All patients were provided written informed consent.

### 2.2. Demographic and Laboratory Data

We recorded demographic data, cardiovascular risk factors, and laboratory data for all patients. Cardiovascular risk factors included smoking status, diabetes mellitus, hypertension, dyslipidemia, peripheral vascular disease, stroke, and history of prior myocardial infarction (MI), PCI, and coronary artery bypass graft surgery. Blood samples were used in the standard battery of hematological and biochemical tests. Serum cystatin C and creatinine were drawn within 24 h before index PCI. Glomerular filtration rate (GFR) was calculated by the MDRD study (Modification of Diet in Renal Disease) equation and chronic kidney disease (CKD) stages were defined according to GFR [11].

### 2.3. PCI Procedure and Follow-Up

All procedures were performed according to the current standard guidelines. The choice of PCI strategy including type of stent, pre-dilatation, post-stenting adjunctive balloon inflation, and the use of intravascular ultrasound or glycoprotein IIb/IIIa inhibitors were all left to the operators’ discretion. All patients were prescribed aspirin daily (100 mg) indefinitely and clopidogrel daily (75 mg) for 1 year, after a loading dose of 300 mg or 600 mg. After index PCI, clinical follow-up was performed at 1, 3, 9, and 12 months and then annually up to 3 years. Repeat angiography was optional at 9 to 13 months. Follow-up data were obtained from outpatient visits or by telephone call or medical questionnaire. For any clinical event, all relevant medical records were reviewed and adjudicated by an external clinical event committee. Cross-checking 100% mortality of patients using a unique personal identification number from the national healthcare system. The mean follow-up duration was 1039 days (interquartile range: 1086 to 1134 days).

### 2.4. Outcomes

The primary outcome was clinical outcomes including major adverse cardiovascular events (MACE) and bleeding events up to 3 years. MACE was defined as the composite of death, MI or repeat revascularization of target vessel. Mortality data were classified into cardiovascular and non-cardiovascular death. Cardiovascular death was defined as death caused by coronary artery disease (CAD), heart failure, arrhythmia, stroke, pulmonary embolism, or other definite vascular causes or unless an undisputed non-cardiovascular cause was present. Non-cardiovascular death was defined as death caused by accidents, cancer, pulmonary disease, and other miscellaneous causes. To verify the accuracy of mortality information, we matched our data to the nationwide official data on death certification offered by the National Statistical Office. MI and stent thrombosis (ST) were defined according to the Academic Research Consortium definitions [12]. The bleeding endpoint was thrombolysis in myocardial infarction (TIMI) major and minor bleeding, during the follow-up duration [13]. The secondary outcomes were all-cause death, MI and target lesion revascularization at 3 years.

### 2.5. Statistical Analyses 

We prespecified patients into two groups: patients in the lowest quartile as Q1, and the remaining group as Q2–4 based on baseline ratio serum creatinine/serum cystatin C, stratified by sex. We used these quartile categories in the following analyses. To test for differences in categorical or continuous variables among quartiles of SI, we used the chi-square test, the one-way analysis of variance test, or the Kruskal-Wallis test. Survival rates after PCI were estimated using the Kaplan-Meier product-limit estimation method, and the survival rates of subjects by SI quartile were compared using log-rank tests.

The Cox proportional hazard model was used to estimate the hazard ratios (HRs) and 95% confidence intervals (CIs) for adverse outcomes among quartiles of SI. The lowest quartile of the SI level was used as a reference. Cox proportional hazards regression model was used with a backward elimination algorithm and 0.05 as the significance level was used. Variables were age, sex, body mass index (BMI), diabetes mellitus, hypertension, peripheral vascular disease, previous MI or PCI, current smoking, previous congestive heart failure, clinical diagnosis, diseased vessel extent and total stent length. For subgroup analysis, the Cox model described above was applied. The cut off values for the exploratory subgroup were set as follows. As for the age, the standard for very elderly people over 75 years old (BMI 25 kg/m^2^) was used, which is the standard for obesity in the Asian Pacific region by World Health Organization; finally, ejection fraction was based on the definition of 40%, defined as a measured ejection fraction of 40% or less, as the cut off value [14]. To assess the additive risk predictive value of SI, four predictive models were created by combining several variables, including, age, SI, and the Synergy between PCI with Taxus and Cardiac Surgery (SYNTAX) score, which is a representation of coronary anatomical complexity. We applied the C-statistics, integrated discrimination index (IDI), and the continuous net reclassification index (NRI) values to compare the reclassification capacities of various models across a continuous range of risk thresholds. All analyses were performed using SPSS software version 25.0 (IBM Corporation, Armonk, NY, USA) and R, version 3.5.2 (R foundation for Statistical Computing, Vienna, Austria).

## 3. Results

### 3.1. Patient Characteristics and Sarcopenia Index

A total of 1086 patients were included in the analysis (mean age, 72.3 ± 5.4), of whom 681 (62.7%) were men, 471 (43.4%) had diabetes mellitus, and 804 (74.0%) had hypertension. SI ranged from 0.39 to 2.39 (median 1.04, interquartile range (IQR) 0.89–1.19). SI was distributed differently depending on sex; the mean SI was 1.11 (IQR 1.00–1.26) in men and 0.89 (IQR 0.78–1.02) in women (*p* < 0.001) (Figure 2). Stratified by sex, the total population was divided into quartiles according to SI and participants were grouped into two groups, a lowest quartile group (Q1) and upper quartile groups (Q2–4).

The demographic, clinical, and angiographic characteristics of patients according to the SI quartile are summarized in Table 1. Briefly, patients in the Q1 group were older and had lower BMI and ejection fraction than the Q2–4 group.

### 3.2. Clinical Outcomes during 3-Year Follow-Up

Among the total population, 3-year MACE occurred in 154 (14.2%) and TIMI major and minor bleeding occurred in 25 (2.3%) patients. Table 2 shows the clinical outcomes according to the two groups during the follow-up period.

Compared to the SI Q2–4 group, SI Q1 group had a significantly higher 3-year MACE rate (Q1 vs. Q2–4; 23.1% vs. 11.2%, *p* < 0.001), while TIMI major and minor bleeding events were similar between two groups (Q1 vs. Q2–4; 2.6% vs. 2.2%, *p* = 0.66). The major contributor of MACE was all-cause death (Q1 vs. Q2–4; 15.8% vs. 4.7%, *p* < 0.001). After adjustment for covariates, SI was a significant predictor of MACE (HR 2.18 (95% CI 1.55–3.06), *p* < 0.001). Kaplan-Meier estimates of clinical events showed that the Q1 group had a significantly higher MACE rate (Figure 3). In subgroup analysis, the poor prognosis of the Q1 group compared to that of the Q2–4 group was consistently observed across the various subgroups, without significant interaction (Figure 4).

### 3.3. Additive Risk Predictive Value of Sarcopenia Index

We next determined whether the addition of SI would improve the prediction of clinical outcomes as MACE. Based on the SYNTAX score, which is a representation of coronary anatomical complexity, we compared the following four predictive models: Model 1, SYNTAX score; Model 2, addition of age to Model 1; Model 3, addition of SI to Model 1; and Model 4, addition of age and SI to Model 1. Measures of diagnostic accuracy including C-statistics, NRI, and IDI for each model are displayed in Table 3. We found a trend toward improvement in the predictive power of clinical models for MACE with the addition of age, SI, and a combination of age and SI (C-statistics 0.49, 0.56, 0.60, and 0.62, respectively). The combination of age and SI in SYNTAX score also significantly improved the prediction of MACE estimated by IDI and NRI (total NRI = 0.102, *p* = 0.020, and category-free IDI = 0.023, *p* = 0.007). 

## 4. Discussion

We evaluated the clinical relevance of SI in older patients with CAD. The main findings were as follows. First, the lowest SI group were older, had lower BMI and depressed left ventricular ejection fractions. Regarding TIMI bleeding events, there were no differences between the two groups (Q1 vs. Q2–4; 2.6% vs. 2.2%, *p* = 0.66). Second, the lowest SI group significantly increased the risk of MACE at 3 years in older patients who underwent PCI with second-generation DES (HR 2.18, 95% CI 1.55–3.06, *p* < 0.001), mainly driven by all-cause death. Third, adding SI could increase the prediction of clinical outcomes including cardiovascular events.

To our knowledge, this is the first study to demonstrate the distribution of SI in patients who have undergone PCI for CAD with second-generation DES and to compare clinical and angiographic findings accordingly. Both univariate and multivariate Cox proportional risk analysis and subgroup analysis demonstrated that the lowest SI is an independent prognostic factor for MACE in the subsequent year. Interestingly, there was no difference in the occurrence of TIMI bleeding events, according to SI. However, because of the small observed bleeding events, it is challenging to assess the magnitude of the significance of our finding.

### 4.1. Sarcopenia Index as a Surrogate for Muscle Mass

Considering the sources of its generation, creatinine comes from skeletal cells and cystatin C comes from all nucleated cells. Serum creatinine is mostly influenced by physiological and clinical conditions that affect muscle mass [15]. Serum creatinine levels are usually low in patients with low muscle mass or sarcopenia. However, cystatin C levels can provide a more useful estimate of kidney function in patients with reduced muscle mass because it is freely filtered from the glomeruli with low molecular weight proteins with stable production rates. Because serum cystatin C values depend on the amount of glomerular filtration, unlike creatinine, they are not affected by premature kidney dysfunction.

The low serum creatinine/serum cystatin C ratio has previously been associated with low muscle mass in various settings [5,16,17]. These studies demonstrated a reliable marker of muscle mass, as measured by DEXA, CT, MRI or electrical bioimpedance. These findings suggest that the creatinine/cystatin ratio eliminated the influence of potential differences in renal function. As recently reviewed, there are many pitfalls in the measurement of muscle mass [18]. These traditional tools are difficult and challenging both to use and to understand; they are expensive, difficult to standardize, and require specific tools. SI measurements are convenient and straightforward to determine. There are reports of the association between SI and muscle mass and poor prognosis in patients with comorbidities [6,17,19]. The current study corroborates these findings.

### 4.2. Association between Low Sarcopenia Index and Prognosis of CAD

Previous studies reported that low muscle mass assessed with various methods was closely associated with subclinical features of CAD, including coronary artery calcification or subclinical atherosclerosis [20,21]. The present study is meaningful because it provides evidence for the association of sarcopenia and poor clinical outcomes in CAD. We wish to emphasize that low SI is an independent predictor of MACE in older patients. Furthermore, combining SI and age provided incremental discrimination of risk for MACE. In daily practice, age or clinical disease-based decision making has been standard practice in patients with CAD [22,23]. However, even in younger patients, there may be sarcopenia or functional decline associated with clinical disease [24]. Measuring SI can provide additional insight into skeletal muscle loss as well as comprehensive indicators associated with them. During long-term follow-up, it is remarkable that an increase in mortality was observed. This would mean that CAD patients with low muscle mass should be considered sufficiently compared to standard-muscle-mass groups. 

The poor prognosis may be due to the sharing of these pathophysiological conditions in the low-muscle-mass group. Sarcopenia and frailty are characterized by the physical function impairment due to their close relationship with the aging process, so that they share overlapping subclinical pathophysiologic conditions [25,26]. In the elderly population with sarcopenia, a slight disruption of the homeostatic balance may lead to the onset of heterogeneous clinical symptoms [27]. We speculated that a healthy adult may easily overcome a stress event (which was PCI in our study), while an elderly with frailty may require a longer recovery period from this stressor and may experience disability. This may be one reason why sarcopenia in our population was a risk for both cardiovascular and non-cardiovascular mortality.

Another possible mechanism for this poor prognosis is to reduce skeletal muscle as a secretory organ and to decrease endocrine function. Myokines are cytokines or other peptides that are produced, expressed and released by skeletal muscle fibers and may contribute in the mediation of the beneficial cardiovascular effects [28]. In patients with low muscle mass, decreased numbers of muscle cells and decrease endocrine function could contribute to poor clinical results. Along with preexisting chronic comorbidities, sarcopenia, frailty, or muscle loss, these factors explain why the mortality is high.

It should be noted that the differences in MACE risk are mainly driven by all-cause death, not by MI and target lesion revascularization. Although patients underwent PCI due to coronary atherosclerosis such as vascular aging, clinical outcomes led to death, which is explained by reduced physiological reserve or increased vulnerability to stressors, not by coronary events [27]. This suggests that, whether frail or not, significant coronary arteriosclerosis can be considered as a target of intervention.

### 4.3. Potential Role of Sarcopenia Index in the Management of CAD

These findings highlight the importance of performing a routine physical assessments for risk stratification and sarcopenia in CAD patients. In this context, sarcopenia is a functional status to be detected early in clinical practice, and the importance of SI is indispensable for identifying simple methods. 

Although the mechanisms underlying the association between low SI and the clinical outcomes of CAD are not fully understood, low muscle mass may be a potential therapeutic target for reducing adverse clinical outcomes in older patients who underwent PCI. A comprehensive multidisciplinary assessment including exercise prescription of rehabilitation strategies should be considered. Exercise-based cardiac rehabilitation reduces cardiovascular mortality and improves long-term outcomes [29,30]. As an indication for cardiac rehabilitation and discussions of muscle mass gains as therapeutic targets, SI could act as a biomarker.

### 4.4. Limitations

This study has several limitations. First, our results were obtained from an observational registry. Although the baseline patient and lesion characteristics were adequately adjusted in the multivariate models, there is a possibility of unobserved confounders. Second, as bleeding events were reported as per the TIMI bleeding classification, we could not capture other bleeds that patients might consider significant, so-called “clinically relevant non-major bleeding events.” In addition, the history of a previous bleeding event, which is recognized as an important risk factor of post-PCI bleeding events, was not collected. Third, we did not evaluate accurate skeletal muscle mass using DEXA, CT, MRI or electrical bioimpedance analyzer. Furthermore, there was no functional evaluation such as muscle strength or physical strength or frailty. Consequently, it is difficult to extend the measured SI to sarcopenia. Therefore, we believe that dedicated confirmatory studies may be needed to confirm the association of sarcopenia or frailty with the SI.

## 5. Conclusions

As a surrogate marker for low muscle mass, the serum creatinine/serum cystatin C ratio is a simple biomarker that is associated with cardiovascular and non-cardiovascular death, but not with bleeding in older patients who underwent PCI.

## Figures and Tables

**Figure 1 jcm-09-03121-f001:**
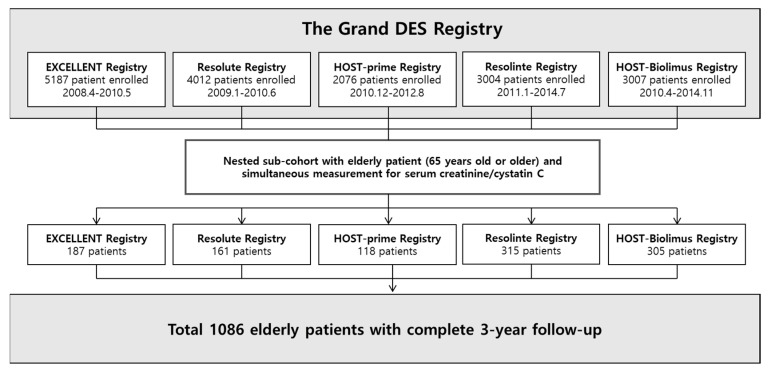
Study flow of the nested sub-cohort. The Grand drug-eluting stents (DES) registry is a multicenter nation-wide pooled registry of the EXCELLENT registry, HOST-PRIME registry, RESOLUTE-Korea registry, RESOLINTE registry, and HOST-BIOLIMUS registry in Korea. Among the total population of the Grand DES registry, 1086 patients who had serum creatinine and cystatin C data and aged 65 or older were analyzed in this study. DES = drug-eluting stent.

**Figure 2 jcm-09-03121-f002:**
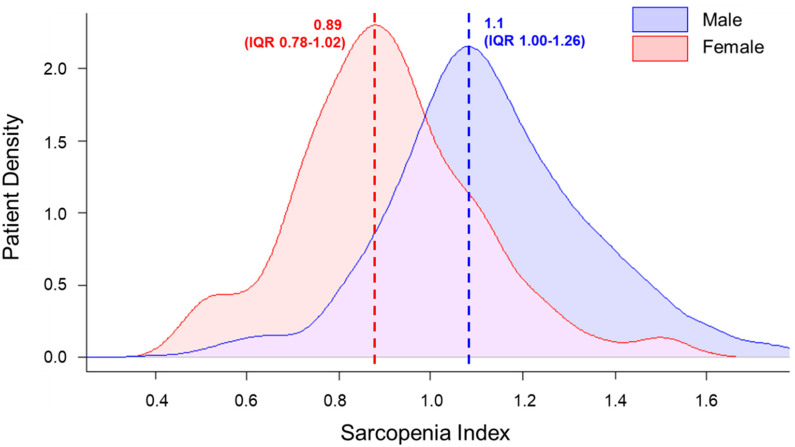
Different distributions of sarcopenia index depending on sex. Sarcopenia index (SI) ranged from 0.39 to 2.39 (median 1.04, interquartile range (IQR) 0.89–1.19). SI was distributed differently depending on sex; the mean SI was 1.11 (IQR 1.00–1.26) in men and 0.89 (IQR 0.78–1.02) in women (*p* < 0.001). SI = sarcopenia index; IQR = interquartile range.

**Figure 3 jcm-09-03121-f003:**
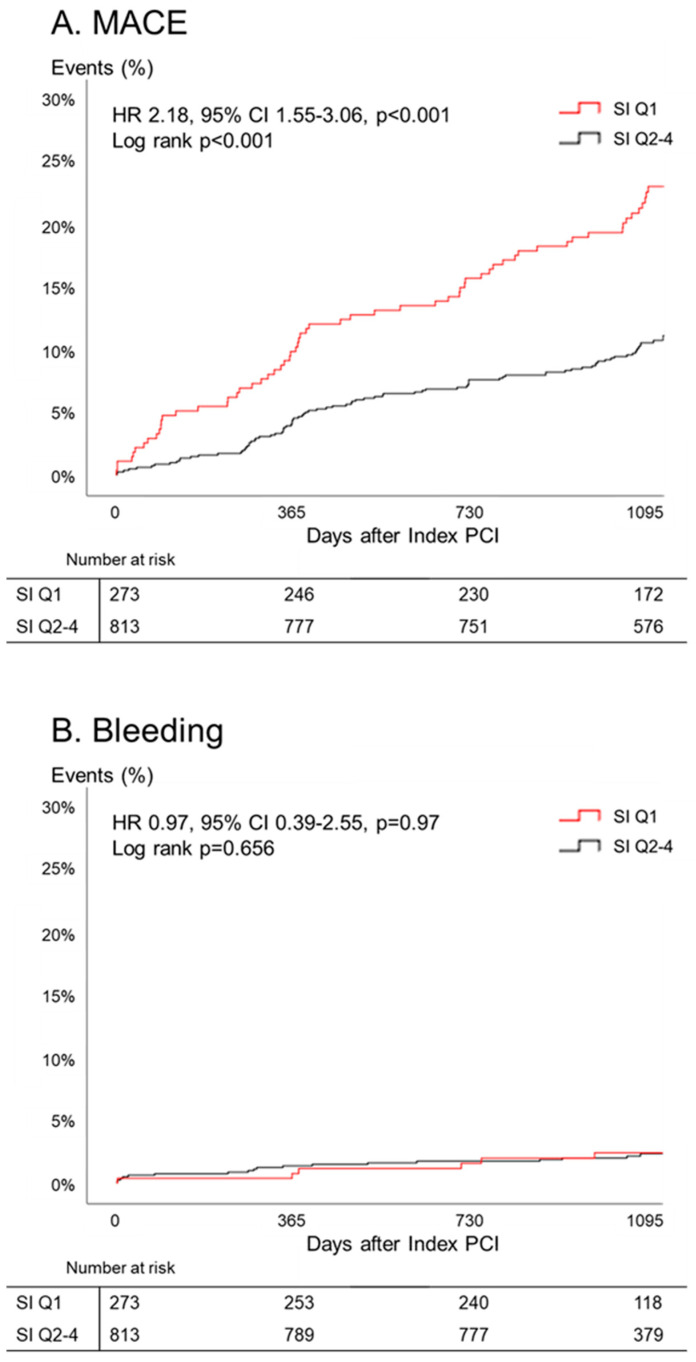
Cumulative incidence of events with quartiles of sarcopenia index. Overall, the lowest quartile group had a higher major advance cardiovascular event (MACE) rate, while there were no significant differences in bleeding events. SI = sarcopenia index; MACE = major advance cardiovascular event; PCI = percutaneous coronary intervention; HR = hazard ratio; CI = confidence interval.

**Figure 4 jcm-09-03121-f004:**
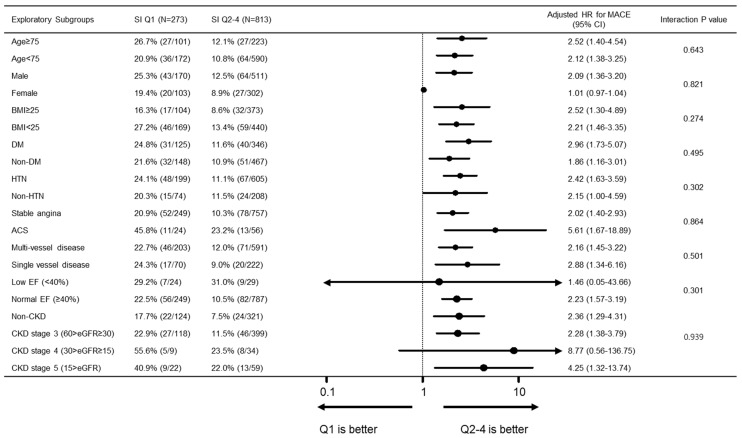
Subgroup analysis for MACE. In subgroup analysis, the poor prognosis of the Q1 group compared to that of the Q2–4 group was consistently observed across the various subgroups, without significant interaction. MACE = major adverse cardiovascular event; BMI = body mass index; DM = diabetes mellitus; HTN = hypertension; ACS = acute coronary syndrome; EF = ejection fraction. The following patient risk factors were included in the multivariate-adjusted Cox proportional hazard regression model: age, sex, body mass index, diabetes mellitus, hypertension, peripheral vascular disease, previous myocardial infarction (MI) or PCI, current smoking, previous congestive heart failure, clinical diagnosis, low EF, diseased vessel extent and total stent length.

**Table 1 jcm-09-03121-t001:** Baseline clinical characteristics of subjects with the lowest quartile (Quartile 1) and upper quartiles (Quartile 2–4) of the SI.

Characteristics	Total (*N* = 1086)	SI Q1 (*N* = 273)	SI Q2–4 (*N* = 813)	*p* Value
SI	1.05 ± 0.24	0.79 ± 0.15	1.13 ± 0.21	<0.001
Age, years	72.3 ± 5.4	73.5 ± 6.1	71.9 ± 5.1	<0.001
BMI, kg/m^2^	24.62 ± 2.95	24.2 ± 3.0	24.7 ± 2.9	0.013
Male	681 (62.7%)	170 (62.3%)	511 (62.9%)	0.863
Diabetes mellitus	471 (43.4%)	125 (45.8%)	346 (42.6%)	0.352
Hypertension	804 (74.0%)	199 (72.9%)	605 (74.4%)	0.620
Dyslipidemia	801 (73.8%)	196 (71.8%)	605 (74.4%)	0.394
Chronic kidney disease	68 (6.3%)	20 (7.3%)	48 (5.9%)	0.401
Prior stroke	116 (10.7%)	28 (10.3%)	88 (10.8%)	0.793
Peripheral vessel disease	40 (3.7%)	13 (4.8%)	27 (3.3%)	0.274
Current smoker	154 (14.2%)	42 (15.4%)	112 (13.8%)	0.510
Prior myocardial infarction	95 (8.8%)	28 (10.3%)	67 (8.2%)	0.308
Prior congestive heart failure	30 (2.8%)	9 (3.3%)	21 (2.6%)	0.534
LVEF, %	59.3 ± 9.7	57.7 ± 12.0	59.8 ± 8.7	0.004
LV dysfunction (EF < 40%)	53 (4.9%)	24 (8.4%)	29 (3.6%)	0.001
Prior percutaneous coronary intervention	211 (19.4%)	54 (19.8%)	157 (19.3%)	0.865
Prior coronary bypass surgery	54 (5.0%)	16 (5.9%)	38 (4.7%)	0.435
Prior revascularization	238 (21.9%)	60 (22.0%)	178 (21.9%)	0.977
Family history of CAD	105 (9.7%)	19 (7.0%)	86 (10.6%)	0.080
Presentations				
Stable angina	743 (68.4%)	167 (61.2%)	576 (70.8%)	0.026
Unstable angina	214 (19.7%)	64 (23.4%)	150 (18.5%)	
NSTEMI	50 (4.6%)	13 (4.8%)	37 (4.6%)	
STEMI	30 (2.8%)	11 (4.0%)	19 (2.3%)	
Silent ischemia	49 (4.5%)	18 (6.6%)	31 (3.8%)	
Angiographic findings				
Extent of angiographic disease				
1VD	292 (26.9%)	70 (25.6%)	222 (27.3%)	0.787
2VD	382 (35.2%)	95 (34.8%)	287 (35.3%)	
3VD	412 (37.9%)	108 (39.6%)	304 (37.4%)	
Left main disease	123 (11.3%)	24 (8.8%)	99 (12.2%)	0.127
At least 1 bifurcation	666 (61.3%)	166 (60.8%)	500 (61.5%)	0.838
At least 1 long lesion	433 (44.3%)	121 (44.3%)	312 (38.4%)	0.083
At least 1 small vessel	605 (55.7%)	163 (59.7%)	442 (54.4%)	0.124
Stent number	1.8 ± 1.1	1.9 ± 1.1	1.8 ± 1.1	0.09
Total stent length, mm	44.7 ± 39.9	47.0 ± 30.1	44.0 ± 29.8	0.141

Data are presented as mean ± standard deviation, or n (%). SI = sarcopenia index; Q = quartile; BMI = body mass index; LVEF = left ventricular ejection fraction; LV = left ventricular; CAD = coronary artery disease; NSTEMI = non-ST-segment elevation myocardial infarction; STEMI = ST-segment elevation myocardial infarction; VD = vessel disease.

**Table 2 jcm-09-03121-t002:** Risk of clinical outcomes in patients with the lowest quartile (Quartile 1) of the SI compared with upper quartiles (Quartile 2–4) of the SI.

Clinical Outcomes	Total (*N* = 1086)	SI Q1 (*N* = 273)	SI Q2–4 (*N* = 813)	Unadjusted HR (95% CI)	*p* Value	Adjusted HR *(95% CI)	*p* Value
MACE	154 (14.2%)	63 (23.1%)	91 (11.2%)	2.23 (1.62–3.07)	<0.001	2.18 (1.55–3.06)	<0.001
All-cause death	81 (7.5%)	43 (15.8%)	38 (4.7%)	3.60 (2.33–5.57)	<0.001	3.48 (2.17–5.58)	<0.001
Cardiovascular death	43 (4.0%)	23 (8.4%)	20 (2.5%)	3.64 (2.00–6.63)	<0.001	3.02 (1.61–5.68)	0.001
Non-cardiovascular death	38 (3.5%)	20 (7.3%)	18 (2.2%)	3.55 (1.88–6.71)	<0.001	3.69 (1.89–7.23)	<0.001
MI	15 (1.4%)	4 (1.5%)	11 (1.4%)	1.15 (0.37–3.61)	0.81	1.16 (0.35-3.81)	0.81
Target lesion revascularization	128 (11.8%)	29 (10.6%)	99 (12.2%)	0.91 (0.60–1.38)	0.67	0.96 (0.63–1.46)	0.96
All bleeding	25 (2.3%)	7 (2.6%)	18 (2.2%)	1.22 (0.51–2.92)	0.66	0.97 (0.37–2.55)	0.97

SI = sarcopenia index; Q = quartile; MACE = major adverse cardiovascular event; MI = myocardial infarction. * The following patient risk factors were included in the multivariate-adjusted Cox proportional hazard regression model: age, sex, body mass index, diabetes mellitus, hypertension, peripheral vascular disease, previous MI or PCI, current smoking, previous congestive heart failure, clinical diagnosis, diseased vessel extent and total stent length.

**Table 3 jcm-09-03121-t003:** Predictive Performance of Models for MACE.

	Model 1. SS	Model 2. SS + Age	Model 3. SS + SI	Model 4. SS + Age + SI
C statistics (95% CI)	0.49 (0.43–0.56)	0.56 (0.49–0.63)	0.60 (0.53–0.68)	0.62 (0.55–0.69)
*p* value	Reference *	<0.001	<0.001	<0.001
NRI (95% CI), Continuous		0.076 (−0.074–0.185)	0.144 (−0.026–0.265)	0.102 (0.015–0.284)
*p* value		0.312	0.106	0.020
IDI (95% CI)		0.010 (0.000–0.043)	0.016 (0.000–0.067)	0.023 (0.001–0.086)
*p* value		0.073	0.060	0.007

* Model 2 through 4 were each compared with model 1. For model comparisons differences in the C statistics, NRI, and IDI, values > 0 indicate better performance for model 1 than the reduced models. SS = SYNTAX score; SI = sarcopenia index; NRI = net reclassification index; IDI = integrated discrimination index.

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
