# Peer review of "Sarcopenia Index as a Predictor of Clinical Outcomes in Older Patients with Coronary Artery Disease"

_jcm, 2020, doi:10.3390/jcm9103121_

Round 1
Reviewer 1 Report
Thank you for the opportunity to read your work.
Although the theme is somewhat interesting, there are some concerns.
[Major concerns]
According to the findings in Table 2, causes of death may be derived from noncardiac death because there was no significant difference of MI or TLR between two groups. Although it might be difficult, causes of death would provide further knowledge regarding SI. Probably sarcopenia would contribute to noncardiac adverse events, rather than cardiac adverse events.
The statement in the Result section 3.3. which explains some models should be described in the Method section.
The reviewer considered that the results of c-statics was somewhat poor. It is natural that the result improves after some variables are added. The reviewer wonders if this analysis is necessary.
In the Discussion section, the authors described that “Because serum cystatin C values depend on the amount of glomerular filtration, unlike creatinine, they are not affected by premature kidney dysfunction”. However, it would be necessary to determine whether the association of SI with the outcomes is observed regardless of renal function. As well as the variables in Figure 4, please conduct sub-analysis concerning CKD stages or renal function.
The authors stated that “Commonly, the presence of sarcopenia is not noticed in comparison with the absolute value of body weight.” The reviewer does not agree with the comment. Please show the grounds.
The authors DOES NOT indicate references in many parts. i.e. The Discussion section 4.2.
The description in the Discussion section “the differences in MACE risk are not mainly driven by all-cause death, not by MI and revascularization of target lesion” is an error according to Table 2.
The authors’ conclusions would be incorrect because there was no difference of cardiovascular events between the groups.
[Minor concerns]
Some definition needs references i.e. CKD, the Academic Research Consortium definitions, etc.
In figure 4, “previous” diabetes mellitus sounds curious. Diabetes mellitus is not cured in many patients.
Author Response
RESPONSE TO REVIEWER 1
[Major concerns]
[Comment 1]
According to the findings in Table 2, causes of death may be derived from noncardiac death because there was no significant difference of MI or TLR between two groups. Although it might be difficult, causes of death would provide further knowledge regarding SI. Probably sarcopenia would contribute to noncardiac adverse events, rather than cardiac adverse events.
[Response]
We appreciate the comments of the reviewer. We agree that it is important to elucidate the differences in clinical outcomes by SI. We stratified the specific cause of mortality into cardiovascular and non-cardiovascular causes, as shown in the following table. The SI Q1 group had a significantly higher risk for both cardiovascular death and non-cardiovascular death, compared to the SI Q2-4 group (Q1 vs. Q2-4; cardiovascular death 8.4% vs. 2.5%, p<0.001; non-cardiovascular death 7.3% vs. 2.2%, p<0.001). After adjustment for covariates, SI was also a significant predictor of cardiovascular death (HR 3.02 [95% CI 1.61-5.68], p=0.001) and non-cardiovascular death (HR 3.69 [95% CI 1.89-7.23], p<0.001).
|
Total (N=1,086) |
SI Q1 (N=273) |
SI Q2-4 (N=813) |
Unadjusted HR (95% CI) |
p value |
Adjusted HR* (95% CI) |
p value |
|
|
MACE |
154 (14.2%) |
63 (23.1%) |
91 (11.2%) |
2.23 (1.62-3.07) |
<0.001 |
2.18 (1.55-3.06) |
<0.001 |
|
All-cause death |
81 (7.5%) |
43 (15.8%) |
38 (4.7%) |
3.60 (2.33-5.57) |
<0.001 |
3.48 (2.17-5.58) |
<0.001 |
|
Cardiovascular death |
43 (4.0%) |
23 (8.4%) |
20 (2.5%) |
3.64 (2.00-6.63) |
<0.001 |
3.02 (1.61-5.68) |
0.001 |
|
Non-cardiovascular death |
38 (3.5%) |
20 (7.3%) |
18 (2.2%) |
3.55 (1.88-6.71) |
<0.001 |
3.69 (1.89-7.23) |
<0.001 |
|
MI |
15 (1.4%) |
4 (1.5%) |
11 (1.4%) |
1.15 (0.37-3.61) |
0.81 |
1.16 (0.35-3.81) |
0.81 |
|
Target lesion revascularization |
128 (11.8%) |
29 (10.6%) |
99 (12.2%) |
0.91 (0.60-1.38) |
0.67 |
0.96 (0.63-1.46) |
0.96 |
|
All bleeding |
25 (2.3%) |
7 (2.6%) |
18 (2.2%) |
1.22 (0.51-2.92) |
0.66 |
0.97 (0.37-2.55) |
0.97 |
Table 2. Risk of clinical outcomes in patients with the lowest quartile (Quartile 1) of the SI compared with upper quartiles (Quartile 2-4) of the SI.
We believe that sarcopenia could be associated with increased risk of both cardiovascular and non-cardiovascular death. In the elderly population with sarcopenia, a slight disruption of the homeostatic balance may lead to the onset of heterogeneous clinical symptoms. We would like to emphasize that a healthy adult may easily overcome a stress event (which was PCI in our study), while an elderly with frailty may need a longer recovery period from this stressor and may experience disability. The following figure illustrates the different response to a stressor in patients with frailty (from Eur J Intern Med. 2016 Nov;35:1-9). Accordingly, this can explain why sarcopenia in our population was a risk for both cardiovascular and non-cardiovascular mortality.
We modified method and discussion section of our manuscript.
Methods - 2. 4. Outcomes
The primary outcome was clinical outcomes including major adverse cardiovascular events (MACE) and bleeding events up to 3 years. MACE was defined as the composite of death, MI or repeat revascularization of target vessel. Mortality data were classified into cardiovascular and non-cardiovascular death. Cardiovascular death was defined as death caused by CAD, heart failure, arrhythmia, stroke, pulmonary embolism, or other definite vascular causes or unless an undisputed non-cardiovascular cause was present. Non-cardiovascular death was defined as death caused by accidents, cancer, pulmonary disease, and other miscellaneous causes.
Discussion - 4.2. Association between low sarcopenia index and prognosis of CAD
The poor prognosis may be due to the sharing of these pathophysiological conditions in the low muscle mass group. Sarcopenia and frailty are characterized by the physical function impairment due to their close relationship with the aging process, so that they share overlapping subclinical pathophysiologic conditions [25,26]. In the elderly population with sarcopenia, a slight disruption of the homeostatic balance may lead to the onset of heterogeneous clinical symptoms [27]. We speculated that a healthy adult may easily overcome a stress event (which was PCI in our study), while an elderly with frailty may require a longer recovery period from this stressor and may experience disability. This may be one reason why sarcopenia in our population was a risk for both cardiovascular and non-cardiovascular mortality.
[Comment 2]
The statement in the Result section 3.3. which explains some models should be described in the Method section.
[Response]
Thank you for this sharp comment. We added the explanation of the model for the additive risk predictive value to the Method section as follows.
Methods - 2. 5. Statistical analyses
To assess the additive risk predictive value of SI, four predictive models were created by combining several variables, including, age, SI, and the Synergy between PCI with Taxus and Cardiac Surgery (SYNTAX) score, which is representation of coronary anatomical complexity. We applied the C-statistics, integrated discrimination index (IDI), and the continuous net reclassification index (NRI) values to compare the reclassification capacities of various models across a continuous range of risk thresholds.
[Comment 3]
The reviewer considered that the results of c-statics was somewhat poor. It is natural that the result improves after some variables are added. The reviewer wonders if this analysis is necessary.
[Response]
We appreciate your valuable comment. In our study, SS has relatively low C-statistics, however the focus of the analysis in Table 3 is the power of the additive risk predictive value of SI. So, Model 2. (SS+Age) and Model 4. (SS+Age+SI) were additionally compared, and this is an additional informative aspect of SI beyond the chronological age in clinical practice. The table is as follows.
|
Model 2. SS + Age |
Model 4. SS + Age + SI |
|
|
C statistics (95% CI) |
0.56 (0.49-0.63) |
0.62 (0.55-0.69) |
|
p value |
Reference |
<0.001 |
|
NRI (95% CI), Continuous |
|
0.256 (0.001-0.511) |
|
p value |
|
0.156 |
|
IDI (95% CI) |
|
0.0179 (0.004-0.0317) |
|
p value |
|
0.0114 |
[Comment 4]
In the Discussion section, the authors described that “Because serum cystatin C values depend on the amount of glomerular filtration, unlike creatinine, they are not affected by premature kidney dysfunction”. However, it would be necessary to determine whether the association of SI with the outcomes is observed regardless of renal function. As well as the variables in Figure 4, please conduct sub-analysis concerning CKD stages or renal function.
[Response]
Thank you for this comment. This is a great suggestion that was well-taken. We agree that additional subgroup analysis is necessary according to GFR. Subgroup analysis was performed according to the CKD stages and according to the presence of CKD, which have been added to the figure follows.
A poor prognosis was consistently observed in the low SI group regardless of GFR with no significant interaction. A relatively wide confidence interval which was observed in the CKD stage 4 group may be explained by the small number of patients (n=43). Collectively, these findings suggest that the SI would be less affected by glomerular filtration.
The final revised figure 4 is as follows.
[Comment 5]
The authors stated that “Commonly, the presence of sarcopenia is not noticed in comparison with the absolute value of body weight.” The reviewer does not agree with the comment. Please show the grounds.
[Response]
Thank you for this comment. We apologize for the possible misleading statement. Sarcopenia usually accompanies weight loss. However, even with the same body weight, the decrease in skeletal muscle or accompanying decline in muscle function may vary. We were trying to stress that it is not just body weight and that the amount of muscle function or muscle decrease needs to be evaluated. We deleted sentences that could be confusing.
[Comment 6]
The authors DOES NOT indicate references in many parts. i.e. The Discussion section 4.2.
[Response]
We have added following references in discussion section of our manuscript.
Discussion - 4.2. Association between low sarcopenia index and prognosis of CAD
Furthermore, combining SI and age provided incremental discrimination of risk for MACE. In daily practice, age or clinical disease-based decision making has been standard practice in patients with CAD [24,25]. However, even in younger patients, there may be sarcopenia or functional decline associated with clinical disease [26]. Measuring SI can provide additional insight into skeletal muscle loss as well as comprehensive indicators associated with them.
The causes and consequences of the specific disease are quite simple in the general adult population, however, the disruption of the homeostatic balance that occurs in the elderly with sarcopenia leads to the onset of heterogeneous clinical symptoms and syndromic states [29]. This may be the basis for explaining that cardiovascular death as well as non-cardiovascular death occurs frequently in the low SI group.
References
- de Araujo Goncalves P, Ferreira J, Aguiar C, Seabra-Gomes R. TIMI, PURSUIT, and GRACE risk scores: sustained prognostic value and interaction with revascularization in NSTE-ACS. Eur Heart J 2005;26:865-72.
- Knuuti J, Wijns W, Saraste A et al. 2019 ESC Guidelines for the diagnosis and management of chronic coronary syndromes. Eur Heart J 2020;41:407-477.
- Rolland Y, Czerwinski S, Abellan Van Kan G et al. Sarcopenia: its assessment, etiology, pathogenesis, consequences and future perspectives. J Nutr Health Aging 2008;12:433-50.
- Cesari M, Nobili A, Vitale G. Frailty and sarcopenia: From theory to clinical implementation and public health relevance. Eur J Intern Med 2016;35:1-9.
[Comment 7]
The description in the Discussion section “the differences in MACE risk are not mainly driven by all-cause death, not by MI and revascularization of target lesion” is an error according to Table 2.
[Response]
We apologize for the mispresentation. We revised the manuscript as follows.
Discussion - 4.2. Association between low sarcopenia index and prognosis of CAD
It should be noted that the differences in MACE risk are not mainly driven by all-cause death, not by MI and target lesion revascularization.
[Comment 8]
The authors’ conclusions would be incorrect because there was no difference of cardiovascular events between the groups.
[Response]
Thank you for your thoughtful comment. This was explained in the response for Comment 1. Although there was no difference in MI or TLR, the risk of both cardiovascular and non-cardiovascular death increased in the low SI group. We observed a significant association between low SI and both cardiovascular and non-cardiovascular mortality. Therefore, we revised conclusions as follows.
Conclusions
As a surrogate marker for low muscle mass, the serum creatinine/serum cystatin C ratio is a simple biomarker that is associated with cardiovascular and non-cardiovascular death, but not with bleeding in older patients who underwent PCI.
[Minor concerns]
[Comment 9]
Some definition needs references i.e. CKD, the Academic Research Consortium definitions, etc.
[Response]
We apologize for the negligence of references. We added following references in discussion section of our manuscript.
Methods - 2. 2. Demographic and laboratory data
Serum cystatin C and creatinine were drawn within 24 hours before index PCI. Glomerular filtration rate (GFR) was calculated by the MDRD study (Modification of Diet in Renal Disease) equation and chronic kidney disease (CKD) stages were defined according to GFR [11].
Methods - 2. 4. Outcomes
MI and stent thrombosis (ST) were defined according to the Academic Research Consortium definitions [12]. The bleeding endpoint was thrombolysis in myocardial infarction (TIMI) major and minor bleeding, during the follow-up duration [13].
References
- Inker LA, Astor BC, Fox CH et al. KDOQI US commentary on the 2012 KDIGO clinical practice guideline for the evaluation and management of CKD. Am J Kidney Dis 2014;63:713-35.
- Garcia-Garcia HM, McFadden EP, Farb A et al. Standardized End Point Definitions for Coronary Intervention Trials: The Academic Research Consortium-2 Consensus Document. Circulation 2018;137:2635-2650.
- Bovill EG, Terrin ML, Stump DC et al. Hemorrhagic events during therapy with recombinant tissue-type plasminogen activator, heparin, and aspirin for acute myocardial infarction. Results of the Thrombolysis in Myocardial Infarction (TIMI), Phase II Trial. Ann Intern Med 1991;115:256-65.
[Comment 10]
In figure 4, “previous” diabetes mellitus sounds curious. Diabetes mellitus is not cured in many patients.
[Response]
Thank you for your sharp comment. We apologize for the misrepresentation. We deleted all corresponding expressions in the manuscript.

Reviewer 2 Report
Here, authors demonstrated the role of Sarcopenia index in older patients with CAD. They used the Grand DES registry which is the combination of the 5 registries to enroll patients, then they follow-up up to 3-year. They found the SI as an independent predictor for CAD. They concluded that SI can predict clinical outcomes including cardiovascular, but not bleeding events in older population with CAD.
I'd like to thank authors. So, SI seems so popular at this time, there is a bunch of papers in the literature about this biomarker, and not only about heart but all systemic diseases. Anyways, please find my comments below;
- Please do not explain Sarcopenia disease, just mention about SI in the introduction part.
- Please do not use words like "nevertheless, etc. " in the scientific papers
- I like your study flow from registries to study population
- Can you explain why didnt you just divide into 4 groups according to quartiles, and what was your purpose putting Q2-Q4 into same group. why didnt you use just cut-off for SI?
- How did you define bleeding?
- Why did you show the first figure? Did you use these interquartile numbers for each sex differently?
- please put all units for each parameter on the table like you did for the "Age, y"!
- Figure 3 and letters are so tiny, could you make them bigger for readers!
- what is the BMI<25 , why did you use 25 as a cut off? What is low EF? Please explain all parameters and cut-off values if you did something with them?
- Please expand methodology section, and explain all procedures you did!
Thanks
Author Response
RESPONSE TO REVIEWER 2
Here, authors demonstrated the role of Sarcopenia index in older patients with CAD. They used the Grand DES registry which is the combination of the 5 registries to enroll patients, then they follow-up up to 3-year. They found the SI as an independent predictor for CAD. They concluded that SI can predict clinical outcomes including cardiovascular, but not bleeding events in older population with CAD.
I'd like to thank authors. So, SI seems so popular at this time, there is a bunch of papers in the literature about this biomarker, and not only about heart but all systemic diseases. Anyways, please find my comments below;
[Comment 1]
Please do not explain Sarcopenia disease, just mention about SI in the introduction part.
[Response]
Thank you for this comment. We modified introduction sections of our manuscript.
Introduction
Sarcopenia is a progressive, generalized skeletal muscle disorder characterized by low muscle strength, low muscle quantity or quality, and low physical performance [1]. The recent increased interest in sarcopenia has prompted efforts to formulate evidence-based clinical guidelines for screening, diagnosis, and management sarcopenia in the older adults [2,3]. There have been many studies of sarcopenia that were applicable not only to patients with coronary artery disease or heart failure but also to patients with cardiac implantable electrical devices or implanted transaortic valves [4-7]. The conventional diagnostic criteria for sarcopenia include both low muscle mass and low muscle function. However, it is not easy to accurately measure muscle mass in routine clinical practice, because measuring the skeletal muscle index requires bioimpedance analysis, computed tomography (CT), magnetic resonance imaging, or dual energy X-ray absorptiometry (DEXA) [2]. By contrast, an indicator that can be easily calculated that reflects skeletal muscle simply is the serum creatinine/serum cystatin C ratio, the so-called sarcopenia index (SI) [3-6].
[Comment 2]
Please do not use words like "nevertheless, etc. " in the scientific papers
[Response]
Thank you for your thoughtful comment. We corrected the all words including “nevertheless” in the manuscript.
[Comment 3]
I like your study flow from registries to study population
[Response]
Thank you for your comment. As was mentioned in the methods section, this study is a nested sub-cohort of the “Grand DES registry”, which includes 5 multicenter registries—HOST-biolimus-3000-Korea, HOST-Excellent-Prime, EXCELLENT prospective cohort, HOST-Resolinte and Resolute-Korea—that enrolled all-comers treated with ≥1 DES without exclusions.
[Comment 4]
Can you explain why didnt you just divide into 4 groups according to quartiles, and what was your purpose putting Q2-Q4 into same group. why didnt you use just cut-off for SI?
[Response]
We appreciate your valuable comment. The analysis according to quartiles was an arbitrary classification. Although there are many studies related to SI, it seems that the consensus of the academic society on SI has not yet been made. For example, Erin F Barreto et al presented cut off value of SI in malnutrition of critically ill patient (JPEN J Parenter Enteral Nutr. 2019 Aug;43(6):780-788). On the other hand, Carlos Antonio Amado et al reported the hospitalization of COPD patients by dividing it into SI quartiles, but unlike our study, this was not stratified by sex (Respiration. 2019;97(4):302-309). It seems clear that low SI has a poor prognosis in several studies with certain patient group, but we think more data should be accumulated to establish a concrete and universal concept of SI. Taken together, we believe that it would be reasonable choice to show SI distribution in patients who underwent PCI, and to classify patients by quartile, rather than the proposal of cutoff values.
[Comment 5]
How did you define bleeding?
[Response]
Bleeding event was defined by thrombolysis in myocardial infarction (TIMI) major and minor bleeding, and we added the reference for the definition as follows. Definition of TIMI major and minor bleeding is as follows. We added following references in discussion section of our manuscript.
- Major
- Any intracranial bleeding (excluding microhemorrhages <10 mm evident only on gradient-echo MRI)
- Clinically overt signs of hemorrhage associated with a drop in hemoglobin of ≥5 g/dL or a ≥15% absolute decrease in hematocrit
- Fatal bleeding (bleeding that directly results in death within 7 days)
- Minor
- Clinically overt (including imaging), resulting in hemoglobin drop of 3 to <5 g/dL or ≥10% decrease in hematocrit
- No observed blood loss: ≥4 g/dL decrease in the hemoglobin concentration or ≥12% decrease in hematocrit
- Any overt sign of hemorrhage that meets one of the following criteria and does not meet criteria for a major or minor bleeding event, as defined above
- Requiring intervention (medical practitioner-guided medical or surgical treatment to stop or treat bleeding, including temporarily or permanently discontinuing or changing the dose of a medication or study drug)
- Leading to or prolonging hospitalization
- Prompting evaluation (leading to an unscheduled visit to a healthcare professional and diagnostic testing, either laboratory or imaging)
Methods - 2. 4. Outcomes
The bleeding endpoint was thrombolysis in myocardial infarction (TIMI) major and minor bleeding, during the follow-up duration [13].
References
- Bovill EG, Terrin ML, Stump DC et al. Hemorrhagic events during therapy with recombinant tissue-type plasminogen activator, heparin, and aspirin for acute myocardial infarction. Results of the Thrombolysis in Myocardial Infarction (TIMI), Phase II Trial. Ann Intern Med 1991;115:256-65.
[Comment 6]
Why did you show the first figure? Did you use these interquartile numbers for each sex differently?
[Response]
We appreciate and respect the opinions of the reviewer. The first figure shows a significantly different distribution of SI according to gender. Therefore, we classified SI, after stratification by gender. This sort of stratification can be also found in GFR calculation. The Cockscroft and Gault formula, MDRD GFR, and CKD-EPI method for GFR uses a different formula according to male or female. Accordingly, we ttried to apply the same rationale in the calculation of SI. The difference in SI according to gender has been observed in other studies as well. (J Int Med Res. 2019 Jul;47(7):3151-3159. Transl Res. 2016 Mar;169:80-90.e1-2)
[Comment 7]
please put all units for each parameter on the table like you did for the "Age, y"!
[Response]
Thank you for your comment. We revised table 1 as follows.
Table 1. Baseline clinical characteristics of subjects with the lowest quartile (Quartile 1) and upper quartiles (Quartile 2-4) of the SI.
|
Total (N=1,086) |
SI Q1 (N=273) |
SI Q2-4 (N=813) |
p value |
|
|
SI |
1.05±0.24 |
0.79±0.15 |
1.13±0.21 |
<0.001 |
|
Age, years |
72.3±5.4 |
73.5±6.1 |
71.9±5.1 |
<0.001 |
|
BMI, kg/m2 |
24.62±2.95 |
24.2±3.0 |
24.7±2.9 |
0.013 |
|
Male |
681 (62.7%) |
170 (62.3%) |
511 (62.9%) |
0.863 |
|
Diabetes mellitus |
471 (43.4%) |
125 (45.8%) |
346 (42.6%) |
0.352 |
|
Hypertension |
804 (74.0%) |
199 (72.9%) |
605 (74.4%) |
0.620 |
|
Dyslipidemia |
801 (73.8%) |
196 (71.8%) |
605 (74.4%) |
0.394 |
|
Chronic kidney disease |
68 (6.3%) |
20 (7.3%) |
48 (5.9%) |
0.401 |
|
Prior stroke |
116 (10.7%) |
28 (10.3%) |
88 (10.8%) |
0.793 |
|
Peripheral vessel disease |
40 (3.7%) |
13 (4.8%) |
27 (3.3%) |
0.274 |
|
Current smoker |
154 (14.2%) |
42 (15.4%) |
112 (13.8%) |
0.510 |
|
Prior myocardial infarction |
95 (8.8%) |
28 (10.3%) |
67 (8.2%) |
0.308 |
|
Prior congestive heart failure |
30 (2.8%) |
9 (3.3%) |
21 (2.6%) |
0.534 |
|
LVEF, % |
59.25±9.66 |
57.69±12.00 |
59.8±8.7 |
0.004 |
|
LV dysfunction (EF < 40%) |
53 (4.9%) |
24 (8.4%) |
29 (3.6%) |
0.001 |
|
Prior percutaneous coronary intervention |
211 (19.4%) |
54 (19.8%) |
157 (19.3%) |
0.865 |
|
Prior coronary bypass surgery |
54 (5.0%) |
16 (5.9%) |
38 (4.7%) |
0.435 |
|
Prior revascularization |
238 (21.9%) |
60 (22.0%) |
178 (21.9%) |
0.977 |
|
Family history of CAD |
105 (9.7%) |
19 (7.0%) |
86 (10.6%) |
0.080 |
|
Presentations |
||||
|
Stable angina |
743 (68.4%) |
167 (61.2%) |
576 (70.8%) |
0.026 |
|
Unstable angina |
214 (19.7%) |
64 (23.4%) |
150 (18.5%) |
|
|
NSTEMI |
50 (4.6%) |
13 (4.8%) |
37 (4.6%) |
|
|
STEMI |
30 (2.8%) |
11 (4.0%) |
19 (2.3%) |
|
|
Silent ischemia |
49 (4.5%) |
18 (6.6%) |
31 (3.8%) |
|
|
Angiographic findings |
||||
|
Extent of angiographic disease |
||||
|
1VD |
292 (26.9%) |
70 (25.6%) |
222 (27.3%) |
0.787 |
|
2VD |
382 (35.2%) |
95 (34.8%) |
287 (35.3%) |
|
|
3VD |
412 (37.9%) |
108 (39.6%) |
304 (37.4%) |
|
|
Left main disease |
123 (11.3%) |
24 (8.8%) |
99 (12.2%) |
0.127 |
|
At least 1 bifurcation |
666 (61.3%) |
166 (60.8%) |
500 (61.5%) |
0.838 |
|
At least 1 long lesion |
433 (44.3%) |
121 (44.3%) |
312 (38.4%) |
0.083 |
|
At least 1 small vessel |
605 (55.7%) |
163 (59.7%) |
442 (54.4%) |
0.124 |
|
Stent number |
1.8±1.1 |
1.9±1.1 |
1.8±1.1 |
0.09 |
|
Total stent length, mm |
44.7±39.9 |
47.0±30.1 |
44.0±29.8 |
0.141 |
Data are presented as mean ± standard deviation, or n (%). SI = sarcopenia index; Q = quartile; BMI = body mass index; LVEF = left ventricular ejection fraction; LV = left ventricular; CAD = coronary artery disease; NSTEMI = non-ST-segment elevation myocardial infarction; STEMI = ST-segment elevation myocardial infarction; VD = vessel disease.
[Comment 8]
Figure 3 and letters are so tiny, could you make them bigger for readers!
[Response]
We apologize for your inconvenience. We revised the figure 3 as follows.
|
|
|
Figure 3. Cumulative incidence of events with quartiles of sarcopenia index. Overall, the lowest quartile group had a higher MACE rate, while there were no significant differences in bleeding events. SI = sarcopenia index; MACE = major advance cardiovascular event; PCI = percutaneous coronary intervention; HR = hazard ratio; CI = confidence interval.
[Comment 9]
what is the BMI<25 , why did you use 25 as a cut off? What is low EF? Please explain all parameters and cut-off values if you did something with them?
[Response]
Thank you for your comment. In subgroup analysis, the evidence for cut off values of EF, BMI, and age are as follows.
First, the cut off value of EF is based on the definition of reduced LV systolic function, which is defined as a measured ejection fraction of 40% or less. Second, the cut off value of BMI was applied based on the ‘World Health Organization. The Asia-Pacific perspective: redefining obesity and its treatment’. In contrast to western, the Asian-Pacific WHO sets the obesity standard of the Asian population at 25 kg/m2. Finally, the cut off value of age was based on the criterion of ‘late elderly’. We added the above mentioned all parameters and cut off values to the methods section and references of our manuscript as follows.
Methods - 2. 5. Statistical analyses
For subgroup analysis, the Cox model described above was applied. The cut off values for the exploratory subgroup were set as follows. As for the age, the standard of late elderly 75 years old, BMI is 25kg/m2, which is the standard for obesity in the Asia Pacific by the World Health Organization, and finally, ejection fraction is based on the definition of 40%, which is a defined as a measured ejection fraction of 40% or less, as the cut off value [14].
References
- World Health Organization. The Asia-Pacific perspective: redefining obesity and its treatment. Sydney: Health Communications Australia; 2000. Regional Office for the Western Pacific.
[Comment 10]
Please expand methodology section, and explain all procedures you did!
[Response]
Thank you for your comment. We described the method section in more detail as follows.
Methods
- 2. Demographic and laboratory data
We recorded demographic data, cardiovascular risk factors, and laboratory data for all patients. Cardiovascular risk factors included smoking status, diabetes mellitus, hypertension, dyslipidemia, peripheral vascular disease, stroke, and history of prior myocardial infarction (MI), PCI, and coronary artery bypass graft surgery. Blood samples were used in the standard battery of hematological and biochemical tests. Serum cystatin C and creatinine were drawn within 24 hours before index PCI. Glomerular filtration rate (GFR) was calculated by the MDRD study (Modification of Diet in Renal Disease) equation and chronic kidney disease (CKD) stages were defined according to GFR [11].
- 3. PCI procedure and follow-up
All procedures were performed according to the current standard guidelines. The choice of PCI strategy including type of stent, pre-dilatation, post-stenting adjunctive balloon inflation, and the use of intravascular ultrasound or glycoprotein IIb/IIIa inhibitors were all left to the operators’ discretion. All patients were prescribed aspirin daily 100mg indefinitely and clopidogrel daily 75mg for 1 year, after a loading dose of 300mg or 600mg. After index PCI, clinical follow-up was performed at 1, 3, 9, and 12 months and then annually up to 3 years. Repeat angiography was optional at 9 to 13 months.
- 4. Outcomes
The primary outcome was clinical outcomes including major adverse cardiovascular events (MACE) and bleeding events up to 3 years. MACE was defined as the composite of death, MI or repeat revascularization of target vessel. Mortality data were classified into cardiovascular and non-cardiovascular death. Cardiovascular death was defined as death caused by coronary artery disease (CAD), heart failure, arrhythmia, stroke, pulmonary embolism, or other definite vascular causes or unless an undisputed non-cardiovascular cause was present. Non-cardiovascular death was defined as death caused by accidents, cancer, pulmonary disease, and other miscellaneous causes. To verify the accuracy of mortality information, we matched our data to the nationwide official data on death certification offered by the National Statistical Office. MI and stent thrombosis (ST) were defined according to the Academic Research Consortium definitions [12]. The bleeding endpoint was thrombolysis in myocardial infarction (TIMI) major and minor bleeding, during the follow-up duration [13]. The secondary outcome was all-cause death, MI and target lesion revascularization at 3 years.
- 5. Statistical analyses
For subgroup analysis, the Cox model described above was applied. The cut off values for the exploratory subgroup were set as follows. As for the age, the standard of late elderly 75 years old, BMI is 25kg/m2, which is the standard for obesity in the Asia Pacific by the World Health Organization, and finally, ejection fraction is based on the definition of 40%, which is a defined as a measured ejection fraction of 40% or less, as the cut off value [14]. To assess the additive risk predictive value of SI, four predictive models were created by combining several variables, including, age, SI, and the Synergy between PCI with Taxus and Cardiac Surgery (SYNTAX) score, which is representation of coronary anatomical complexity. We applied the C-statistics, integrated discrimination index (IDI), and the continuous net reclassification index (NRI) values to compare the reclassification capacities of various models across a continuous range of risk thresholds.

Reviewer 3 Report
In the manuscript “Sarcopenia index as a predictor of clinical outcomes in older patients with coronary artery disease”, the authors have analyzed the impact of sarcopenia on the outcome of patients with coronary heart disease. The topic of the manuscript is certainly interesting.
They have found that the Sarcopenia Index is a surrogate for the degree of muscle mass, a useful tool to identify high-risk patients and it is an independent predictor for MACE events.
The manuscript is well written and the analysis is appropriate.
However, there are some points that need further clarification:
- Why did the authors consider patients over 65 years of age? Please explain why they chose this age threshold.
- In Table 1 there is no information about previous bleedings. Since bleeding is a considered endpoint, it would be useful to know if patients have had other bleeding events and insert this variable in the model.
- Could a model that also adjusts for ventricular dysfunction be more correct?
- There seem to be some errors in the calculation of percentages. In particular reference is made to Table 1 (e.g. LVEF dysfunction 23/273 = 8.4% and not 9.9% as reported). Please recheck all the percentage calculations.
Author Response
REVIEWER 3
In the manuscript “Sarcopenia index as a predictor of clinical outcomes in older patients with coronary artery disease”, the authors have analyzed the impact of sarcopenia on the outcome of patients with coronary heart disease. The topic of the manuscript is certainly interesting.
They have found that the Sarcopenia Index is a surrogate for the degree of muscle mass, a useful tool to identify high-risk patients and it is an independent predictor for MACE events. The manuscript is well written and the analysis is appropriate.
However, there are some points that need further clarification:
[Comment 1]
Why did the authors consider patients over 65 years of age? Please explain why they chose this age threshold.
[Response]
We appreciate your valuable comment. Sarcopenia is a syndrome characterized by low muscle strength, low muscle quantity or quality, and low physical performance. Although the development of sarcopenia is now recognized to begin earlier in life, research into the etiology of sarcopenia has focused on the adult determinants of muscle loss in older people. Even in the current guidelines, sarcopenia is classified as primary (aging) and secondary (illness, inactivity, malnutrition). Most of the previous studies of sarcopenia described the study population patients over 65 years of age, so-called 'older adults', 'older people', or 'older patients aged 65 years or older.' That's why we set the age threshold for SI studies at 65.
[Comment 2]
In Table 1 there is no information about previous bleedings. Since bleeding is a considered endpoint, it would be useful to know if patients have had other bleeding events and insert this variable in the model.
[Response]
We understand that the information about previous bleeding events are of great importance. According to a recently published white paper from the Academic Research Consortium (ARC), a history of recent spontaneous bleeding event or previous spontaneous intracranial hemorrhage was defined as a major determinant of ‘high bleeding risk’. Unfortunately however, data of the individual patient's previous bleeding event was not collected. We added this as a limitation in our manuscript marked in red, as follows.
Discussion – 4.4 Limitations
Second, as bleeding events were reported as per the TIMI bleeding classification, we could not capture other bleeds that patients might consider significant, so-called “clinically relevant non-major bleeding events.” Also, the history of a previous bleeding event, which is recognized as an important risk factor of post-PCI bleeding events, was not collected.
[Comment 3]
Could a model that also adjusts for ventricular dysfunction be more correct?
[Response]
Thank you for your valuable comment.
The mean EF as a continuous variable between Q1 and Q2-4 is different statistically. However, both mean values are within the normal EF range and thus the difference does not appear to be clinically significant (Q1 vs. Q2-4, 57.7±12.0 vs. 59.8±8.7, p=0.001).
We set the cut off value of EF based on the definition of reduced LV systolic function, which is defined as a measured ejection fraction of 40% or less.
This variable was adjusted to the Cox proportional hazard model, however, reduced EF itself did not act as an independent predictor for MACE (adjusted HR 1.20, 95% CI 0.66-2.17, p=0.556). The following patient risk factors were included in the multivariate-adjusted Cox proportional hazard regression model: age, sex, body mass index, diabetes mellitus, hypertension, peripheral vascular disease, previous MI or PCI, current smoking, previous congestive heart failure, clinical diagnosis, low EF, diseased vessel extent and total stent length. On the other hand, in the subgroup analysis of patients with reduced EF (<40%), a relatively wide confidence interval was observed, which seems to be caused by a small number of samples (n=53).
[Comment 4]
There seem to be some errors in the calculation of percentages. In particular reference is made to Table 1 (e.g. LVEF dysfunction 23/273 = 8.4% and not 9.9% as reported). Please recheck all the percentage calculations.
[Response]
Thank you for your sharp comment. We sincerely apologize for the confusion caused by the incorrect notation. We revised Table 1. in the manuscript as follows.
|
Total (N=1,086) |
SI Q1 (N=273) |
SI Q2-4 (N=813) |
p value |
|
|
LVEF, % |
59.3±9.7 |
57.7±12.0 |
59.8±8.7 |
0.004 |
|
LV dysfunction (EF < 40%) |
53 (4.9%) |
24 (8.4%) |
29 (3.6%) |
0.001 |
Round 2
Reviewer 1 Report
Dear authors,
Thank you for your polite response and your manuscript has improved.
Reviewer 2 Report
Thank you for submitting your carefully-prepared revision, which satisfactorily addressed the remaining concerns.
Reviewer 3 Report
The authors have responded to the above comments. The manuscript seems to have improved. I have no further comments.